# ADVERSARIAL EXPLORATION STRATEGY FOR SELF-SUPERVISED IMITATION LEARNING

## ABSTRACT

We present an adversarial exploration strategy, a simple yet effective imitation learning scheme that incentivizes exploration of an environment without any extrinsic reward or human demonstration. Our framework consists of a deep reinforcement learning (DRL) agent and an inverse dynamics model contesting with each other. The former collects training samples for the latter, and its objective is to maximize the error of the latter. The latter is trained with samples collected by the former, and generates rewards for the former when it fails to predict the actual action taken by the former. In such a competitive setting, the DRL agent learns to generate samples that the inverse dynamics model fails to predict correctly, and the inverse dynamics model learns to adapt to the challenging samples. We further propose a reward structure that ensures the DRL agent collects only moderately hard samples and not overly hard ones that prevent the inverse model from imitating effectively. We evaluate the effectiveness of our method on several OpenAI gym robotic arm and hand manipulation tasks against a number of baseline models. Experimental results show that our method is comparable to that directly trained with expert demonstrations, and superior to the other baselines even without any human priors.

## 1 INTRODUCTION

Over the past decade, imitation learning (IL) has been successfully applied to a wide range of domains, including robot learning (Englert et al., 2013; Schulman et al., 2013), autonomous navigation (Choudhury et al., 2017; Ross et al., 2013), manipulation tasks (Nair et al., 2017; Prieur et al., 2012), and self-driving cars (Codevilla et al., 2018). Traditionally, IL aims to train an imitator to learn a control policy $\pi$ only from expert demonstrations. The imitator is typically presented with multiple demonstrations during the training phase, with an aim to distill them into $\pi$. To learn $\pi$ effectively and efficiently, a large set of high-quality demonstrations are necessary. This is especially prevalent in current state-of-the-art IL algorithms, such as dataset aggregation (DAgger) (Ross et al., 2011) and generative adversarial imitation learning (GAIL) (Ho & Ermon, 2016). Although these approaches have been the dominant algorithms in IL, a major bottleneck for them is their reliance on high-quality demonstrations, which often require extensive supervision from human experts. In addition, a serious flaw in the learned policy $\pi$ is its tendency to overfit to demonstration data, preventing it from generalizing to new ones. To overcome the aforementioned challenges in IL, a number of methods have been investigated to enhance the generalizability and data efficiency, or reduce the degree of human supervision. Initial efforts in this direction were based on the idea of meta learning (Duan et al., 2017; Finn et al., 2017; Yu et al., 2018), in which the imitator is trained from a meta learner that is able to quickly learn a new task with only a few set of demonstrations. However, such schemes still require training the meta-learner with tremendous amount of time and demonstration data, leaving much room for improvement. Thus, a rapidly-growing body of literature based on the concept of using forward/inverse dynamics models to learn $\pi$ within an environment in a self-supervised fashion (Agrawal et al., 2016; Nair et al., 2017; Pathak et al., 2018) has emerged in the past few years. One key advantage of the concept is that it provides an autonomous way for preparing training data, removing the need of human intervention. In this paper, we call it *self-supervised IL*.

Self-supervised IL allows an imitator to collect training data by itself instead of using predefined extrinsic reward functions or expert supervision during training. It only needs demonstration during inference, drastically decreasing the time and effort required from human experts. Although the core principles of self-supervised IL are straightforward and have been exploited in many fields (Agrawal et al., 2016; Nair et al., 2017; Pathak et al., 2017; 2018), recent research efforts have been dedicated

to addressing the challenges of multi-modality and multi-step planning. For example, the use of forward consistency loss and forward regularizer have been extensively investigated to enhance the task performance of the imitator (Agrawal et al., 2016; Pathak et al., 2018). This becomes especially essential when the lengths of trajectories grow and demonstration samples are sparse, as multiple paths may co-exist to lead the imitator from its initial observation to the goal observation. The issue of multi-step planning has also drawn a lot of attention from researchers, and is usually tackled by recurrent neural networks (RNNs) and step-by-step demonstrations (Nair et al., 2017; Pathak et al., 2018). The above self-supervised IL approaches report promising results, however, most of them are limited in applicability due to several drawbacks. First, traditional methods of data collection are usually inefficient and time-consuming. Inefficient data collection results in poor exploration, giving rise to a degradation in robustness to varying environmental conditions (e.g., noise in motor control) and generalizability to difficult tasks. Second, human bias in data sampling range tailored to specific interesting configurations is often employed (Agrawal et al., 2016; Nair et al., 2017). Although a more general exploration strategy called curiosity-driven exploration was later proposed in Pathak et al. (2017), it focuses only on exploration in states novel to the forward dynamics model, rather than those directly influential to the inverse dynamics model. Furthermore, it does not discuss the applicability to continuous control domains, and fails in high dimensional action spaces according to our experiments in Section 4. Unlike the approaches discussed above, we do not propose to deal with multi-modality or multi-step planning. Instead, we focus our attention on improving the overall quality of the collected samples in the context of self-supervised IL. This motivates us to equip the model with the necessary knowledge to explore the environment in an efficient and effective fashion.

In this paper, we propose a straightforward and efficient self-supervised IL scheme, called **adversarial exploration strategy**, which motivates exploration of an environment in a self-supervised manner (i.e., without any extrinsic reward or human demonstration). Inspired by Pinto et al. (2017); Shioya et al. (2018); Sukhbaatar et al. (2018), we implement the proposed strategy by jointly training a deep reinforcement learning (DRL) agent and an inverse dynamics model competing with each other. The former explores the environment to collect training data for the latter, and receives rewards from the latter if the data samples are considered difficult. The latter is trained with the training data collected by the former, and only generates rewards when it fails to predict the true actions performed by the former. In such an adversarial setting, the DRL agent is rewarded only for the failure of the inverse dynamics model. Therefore, the DRL agent learns to sample hard examples to maximize the chances to fail the inverse dynamics model. On the other hand, the inverse dynamics model learns to be robust to the hard examples collected by the DRL agent by minimizing the probability of failures. As a result, as the inverse dynamics model becomes stronger, the DRL agent is also incentivized to search for harder examples to obtain rewards. Overly hard examples, however, may lead to biased exploration and cause instability of the learning process. In order to stabilize the learning curve of the inverse dynamics model, we further propose a reward structure such that the DRL agent is encouraged to explore moderately hard examples for the inverse dynamics model, but refraining from too difficult ones for the latter to learn. The self-regulating feedback structure between the DRL agent and the inverse dynamics model enables them to automatically construct a curriculum for exploration.

We perform extensive experiments to validate adversarial exploration strategy on multiple OpenAI gym (Brockman et al., 2016) robotic arm and hand manipulation task environments simulated by the MuJoCo physics engine (Todorov et al., 2012), including *FetchReach*, *FetchPush*, *FetchPickAndPlace*, *FetchSlide*, and *HandReach*. These environments are intentionally selected by us for evaluating the performance of inverse dynamics model, as each of them allows only a very limited set of chained actions to transition the robotic arms and hands to target observations. We examine the effectiveness of our method by comparing it against a number of self-supervised IL schemes. The experimental results show that our method is more effective and data-efficient than the other self-supervised IL schemes for both low- and high-dimensional observation spaces, as well as in environments with high-dimensional action spaces. We also demonstrate that in most of the cases the performance of the inverse dynamics model trained by our method is comparable to that directly trained with expert demonstrations. The above observations suggest that our method is superior to the other self-supervised IL schemes even in the absence of human priors. We further evaluate our method on environments with action space perturbations, and show that our method is able to achieve satisfactory success rates. To justify each of our design decisions, we provide a comprehensive set of ablative analysis and discuss their implications. The contributions of this work are summarized as follows:

- We introduce an adversarial exploration strategy for self-supervised IL. It consists of a DRL agent and an inverse dynamics model developed for efficient exploration and data collection.

- We employ a competitive scheme for the DRL agent and the inverse dynamics model, enabling them to automatically construct a curriculum for exploration of observation space.
- We introduce a reward structure for the proposed scheme to stabilize the training process.
- We demonstrate the proposed method and compare it with a number of baselines for multiple robotic arm and hand manipulation tasks in both low- and high-dimensional state spaces.
- We validate that our method is generalizable to tasks with high-dimensional action spaces.

The remainder of this paper is organized as follows. Section 2 introduces background material. Section 3 describes the proposed adversarial exploration strategy in detail. Section 4 reports the experimental results, and provides an in-depth ablative analysis of our method. Section 5 concludes.

## 2 BACKGROUND

In this section, we briefly review DRL, policy gradient methods, as well as inverse dynamics model.

### 2.1 DEEP REINFORCEMENT LEARNING AND POLICY GRADIENT METHODS

DRL trains an agent to interact with an environment $\mathcal{E}$. At each timestep $t$, the agent receives an observation $x_t \in \mathcal{X}$, where $\mathcal{X}$ is the observation space of $\mathcal{E}$. It then takes an action $a_t$ from the action space $\mathcal{A}$ based on its current policy $\pi$, receives a reward $r$, and transitions to the next observation $x'$. The policy $\pi$ is represented by a deep neural network with parameters $\theta$, and is expressed as $\pi(a|x, \theta)$. The goal of the agent is to learn a policy to maximize the discounted sum of rewards $G_t$:

$$G_t = \sum_{\tau=t}^{T} \gamma^{\tau-t} r(x_\tau, a_\tau), \tag{1}$$

where $t$ is the current timestep, $\gamma \in (0, 1]$ the discount factor, and $T$ the horizon. Policy gradient methods (Mnih et al., 2016; Sutton et al., 2000; Williams, 1992) are a class of RL techniques that directly optimize the parameters of a stochastic policy approximator using policy gradients. Although these methods have achieved remarkable success in a variety of domains, the high variance of gradient estimates has been a major challenge. Trust region policy optimization (TRPO) (Schulman et al., 2015) circumvented this problem by applying a trust-region constraint to the scale of policy updates. However, TRPO is a second-order algorithm, which is relatively complicated and not compatible with architectures that embrace noise or parameter sharing (Schulman et al., 2017). In this paper, we employ a more recent family of policy gradient methods, called proximal policy optimization (PPO) (Schulman et al., 2017). PPO is an approximation to TRPO, which similarly prevents large changes to the policy between updates, but requires only first-order optimization. PPO is superior in its generalizability and sample complexity while retaining the stability and reliability of TRPO [1].

### 2.2 INVERSE DYNAMICS MODEL

An inverse dynamics model $I$ takes as input a pair of observations $(x, x')$, and predicts the action $\hat{a}$ required to reach the next observation $x'$ from the current observation $x$. It is formally expressed as:

$$\hat{a} = I(x, x'|\theta_I), \tag{2}$$

where $(x, x')$ are sampled from the collected data, and $\theta_I$ represents the trainable parameters of $I$. During the training phase, $\theta_I$ is iteratively updated to minimize the loss function $L_I$, expressed as:

$$L_I(a, \hat{a}|\theta_I) = d(a, \hat{a}), \tag{3}$$

where $d$ is a distance metric, and $a$ the ground truth action. During the testing phase, a sequence of observations $\{\hat{x}_0, \hat{x}_1, \cdots, \hat{x}_T\}$ is first captured from an expert demonstration. A pair of observations $(\hat{x}_t, \hat{x}_{t+1})$ is then fed into $I$ at each timestep $t$. Starting from $\hat{x}_0$, the objective of $I$ is to predict a sequence of actions $\{\hat{a}_0, \hat{a}_1, \cdots, \hat{a}_{T-1}\}$ and transition the final observation $\hat{x}_T$ as close as possible.

## 3 METHODOLOGY

In this section, we first describe the proposed adversarial exploration strategy. We then explain the training methodology in detail. Finally, we discuss a technique for stabilizing the training process.

### 3.1 ADVERSARIAL EXPLORATION STRATEGY

Fig. 1 shows a framework that illustrates the proposed adversarial exploration strategy, which includes a DRL agent $P$ and an inverse dynamics model $I$. Assume that $\Phi_\pi : \{x_0, a_0, x_1, a_1 \cdots, x_T\}$ is the

---

[1]For more details on PPO, please refer to supplementary material S.2.

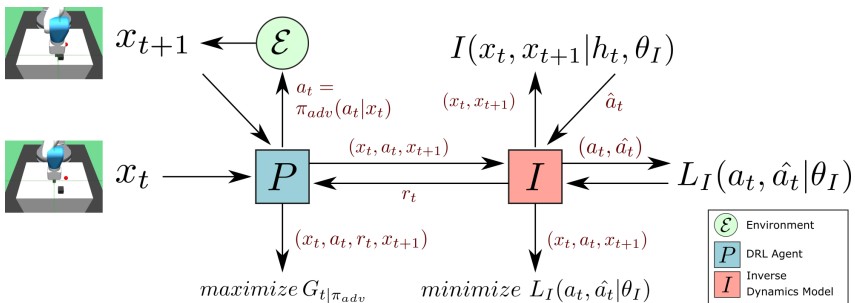

Figure 1: Framework of adversarial exploration strategy.

sequence of observations and actions generated by $P$ as it explores $\mathcal{E}$ using a policy $\pi$. At each timestep $t$, $P$ collects a 3-tuple training sample $(x_t, a_t, x_{t+1})$ for $I$, while $I$ predicts an action $\hat{a}_t$ and generates a reward $r_t$ for $P$. In this work, $I$ is modified from Eq. (2) to include an additional hidden vector $h_t$, which recurrently encodes the information of the past observations. $I$ is thus expressed as:

$$\hat{a}_t = I(x_t, x_{t+1}|h_t, \theta_I)$$
$$h_t = f(h_{t-1}, x_t), \tag{4}$$

where $f(\cdot)$ denotes the recurrent function. $\theta_I$ is iteratively updated to minimize $L_I$, formulated as:

$$\min_{\theta_I} L_I(a_t, \hat{a}_t|\theta_I) = \min_{\theta_I} \beta||a_t - \hat{a}_t||^2, \tag{5}$$

where $\beta$ is a scaling constant. We employ mean squared error $\beta||a_t - \hat{a}_t||^2$ as the distance metric $d(a_t, \hat{a}_t)$, since we only consider continuous control domains in this paper. It can be replaced with a cross-entropy loss for discrete control tasks. We directly use $L_I$ as the reward $r_t$ for $P$, expressed as:

$$r_t(x_t, a_t, x_{t+1}) = L_I(a_t, \hat{a}_t|\theta_I) = \beta||a_t - I(x_t, x_{t+1}|h_t, \theta_I)||^2. \tag{6}$$

Our method targets at improving both the quality and efficiency of the data collection process performed by $P$, as well as the performance of $I$. Therefore, the goal of the proposed framework is twofold. First, $P$ has to learn an adversarial policy $\pi_{adv}(a_t|x_t)$ such that its cumulated discounted rewards $G_{t|\pi_{adv}} = \sum_{\tau=t}^{T} \gamma^{\tau-t} r_t(x_\tau, a_\tau, x_{\tau+1})$ is maximized. Second, $I$ requires to learn an optimal $\theta_I$ such that Eq. (6) is minimized. Minimizing $L_I$ (i.e., $r_t$) leads to decreased $G_{t|\pi_{adv}}$, forcing $P$ to enhance $\pi_{adv}$ to explore more difficult samples to increase $G_{t|\pi_{adv}}$. This implies that $P$ is motivated to focus on $I$'s weak points, instead of randomly collecting ineffective training samples. Training $I$ with hard samples not only accelerates its learning progress, but also helps to boost its performance.

### 3.2 TRAINING METHODOLOGY

We describe the training methodology of our adversarial exploration strategy by a pseudocode presented in Algorithm 1. Assume that $P$'s policy $\pi_{adv}$ is parameterized by a set of trainable parameters $\theta_P$, and is represented as $\pi_{adv}(a_t|x_t, \theta_P)$. We create two buffers $Z_P$ and $Z_I$ for storing the training samples of $P$ and $I$, respectively. In the beginning, $Z_P, Z_I, \mathcal{E}, \theta_P, \theta_I, \pi_{adv}$, as well as a timestep cumulative counter $c$ are initialized. A number of hyperparameters are set to appropriate values, including the number of iterations $N_{iter}$, the number of episodes $N_{episode}$, the horizon $T$, as well as the update period $\mathcal{T}_P$ of $\theta_P$. At each timestep $t$, $P$ perceives the current observation $x_t$ from $\mathcal{E}$, takes an action $a_t$ according to $\pi_{adv}(a_t|x_t, \theta_P)$, and receives the next observation $x_{t+1}$ and a termination indicator $\xi$ (lines 9-11). $\xi$ is set to 1 only when $t$ equals $T$, otherwise it is set to 0. We then store $(x_t, a_t, x_{t+1}, \xi)$ and $(x_t, a_t, x_{t+1})$ in $Z_P$ and $Z_I$, respectively. We update $\theta_P$ every $\mathcal{T}_P$ timesteps using the samples stored in $Z_P$, as shown in (lines 13-21). At the end of each episode, we update $\theta_I$ with samples drawn from $Z_I$ according to the loss function $L_I$ defined in Eq. (5) (line 23).

### 3.3 STABILIZATION TECHNIQUE

Although adversarial exploration strategy is effective in collecting hard samples, it requires additional adjustments if $P$ becomes too strong such that the collected samples are too difficult for $I$ to learn. Overly difficult samples lead to a large variance in gradients derived from $L_I$, which in turn cause a performance drop in $I$ and instability in its learning process. We analyze this phenomenon in greater detail in Section 4.5. To tackle the issue, we propose a training technique that reshapes $r_t$ as follows:

$$r_t := -|r_t - \delta|, \tag{7}$$

---

**Algorithm 1** Adversarial exploration strategy

1: Initialize $Z_P$, $Z_I$, $\mathcal{E}$, and model parameters $\theta_P$ & $\theta_I$
2: Initialize $\pi_{adv}(a_t | x_t, \theta_P)$
3: Initialize the timestep cumulative counter $c = 0$
4: Set $N_{iter}$, $N_{episode}$, $T$, and $\mathcal{T}_P$
5: **for** iteration $i = 1$ to $N_{iter}$ **do**
6:    **for** episode $e = 1$ to $N_{episode}$ **do**
7:       **for** timestep $t = 0$ to $T$ **do**
8:          $P$ perceives $x_t$ from $\mathcal{E}$, and predicts an action $a_t$ according to $\pi_{adv}(a_t | x_t, \theta_P)$
9:          $x_{t+1} = \mathcal{E}(x_t, a_t)$
10:          $\xi = \mathbb{1}[t == T]$
11:          Store $(x_t, a_t, x_{t+1}, \xi)$ in $Z_P$
12:          Store $(x_t, a_t, x_{t+1})$ in $Z_I$
13:          **if** $(c \% \mathcal{T}_P) == 0$ **then**
14:             Initialize an empty batch $B$
15:             Initialize a recurrent state $h_t$
16:             **for** $(x_t, a_t, x_{t+1}, \xi)$ in $Z_P$ **do**
17:                Evaluate $\hat{a}_t = I(x_t, x_{t+1} | h_t, \theta_I)$ (calculated from Eq. (4))
18:                Evaluate $r_t(x_t, a_t, x_{t+1}) = L_I(a_t, \hat{a}_t | \theta_I)$ (calculated from Eq. (6))
19:                Store $(x_t, a_t, x_{t+1}, r_t)$ in $B$
20:             Update $\theta_P$ with the gradient calculated from the samples of $B$
21:             Reset $Z_P$
22:          $c = c + 1$
23:    Update $\theta_I$ with the gradient calculated from the samples of $Z_I$ (according to Eq. (5))
24: **end**

---

where $\delta$ is a pre-defined threshold value. This technique poses a restriction on the range of $r_t$, driving $P$ to gather moderate samples instead of overly hard ones. Note that the value of $\delta$ affects the learning speed and the final performance. We plot the impact of $\delta$ on the learning curve of $I$ in Section 4.5. We further provide an example in our supplementary material to visualize the effect of this technique.

## 4 EXPERIMENTAL RESULTS

In this section, we present experimental results for a series of robotic tasks, and validate that (i) our method is effective in both low- and high-dimensional observation spaces; (ii) our method is effective in environments with high-dimensional action spaces; (iii) our method is more data efficient than the baseline methods; and (iv) our method is robust against action space perturbations. We first introduce our experimental setup. Then, we report experimental results of robotic arm and hand manipulation tasks. Finally, we present a comprehensive set of ablative analysis to validate our design decisions.

### 4.1 EXPERIMENTAL SETUP

We first describe the environments and tasks. Next, we explain the evaluation procedure and the method for collecting expert demonstrations. We then walk through the baselines used for comparison.

#### 4.1.1 ENVIRONMENTS AND TASKS

We evaluate our method on a number of robotic arm and hand manipulation tasks via OpenAI gym (Brockman et al., 2016) environments simulated by the MuJoCo (Todorov et al., 2012) physics engine. We use the Fetch and Shadow Dexterous Hand (Plappert et al., 2018b) for the arm and hand manipulation tasks, respectively. For the arm manipulation tasks, which include *FetchReach*, *FetchPush*, *FetchPickAndPlace*, and *FetchSlide*, the imitator (i.e., the inverse dynamic model $I$) takes as inputs the positions and velocities of a gripper and a target object. It then infers the gripper's action in 3-dimensional space to manipulate it. For the hand manipulation task *HandReach*, the imitator takes as inputs the positions and velocities of the fingers of a robotic hand, and determines the velocities of the joints to achieve the goal. In addition to low-dimensional observations (i.e., position, velocity, and gripper state), we further perform experiments for the above tasks using visual observations (i.e., high-dimensional observations) in the form of camera images taken from a third-person perspective. The detailed description of the above tasks is specified in Plappert et al. (2018b). For the detailed configurations of these tasks, please refer to our supplementary material.

#### 4.1.2 EVALUATION PROCEDURE

The primary objective of our experiments is to demonstrate the efficiency of the proposed adversarial exploration strategy in collecting training data (in a self-supervised manner) for the imitator. We compare our strategy against a number of self-supervised data collection methods (referred to as "baselines" or "baseline methods") described in Section 4.1.4. As different baseline methods employ different data collection strategies, the learning curve of the imitator also varies for different cases. For a fair comparison, the model architecture of the imitator and the amount of training data are fixed

for all cases. All of the experimental results are evaluated and averaged over 20 trials, corresponding to 20 different random initial seeds. In each trial, we train an imitator by the training data collected by a single self-supervised data collection method. At the beginning of each episode, the imitator receives a sequence of observations $\{\hat{x}_0, \hat{x}_1, \cdots, \hat{x}_T\}$ from a successful expert demonstration. At each timestep $t$, the imitator infers an action $\hat{a}_t$ from an expert observation $\hat{x}_{t+1}$ and its current observation $x_t$ by Eq. (4). We periodically evaluate the imitator every 10K timesteps. The evaluation is performed by averaging the success rates of reaching $\hat{x}_T$ over 500 episodes. The configuration of the imitator and the hyperparameters of the baselines are summarized in the supplementary material.

### 4.1.3 COLLECTION OF EXPERT DEMONSTRATIONS

For each task mentioned in Section 4.1.1, we first randomly configure task-relevant settings (e.g., goal position, initial state, etc.). We then collect demonstrations from non-trivial and successful episodes performed by a pre-trained expert agent (Andrychowicz et al., 2017). Please note that the collected demonstrations only contain sequences of observations. The implementation details of the expert agent and the method for filtering out trivial episodes are presented in our supplementary material.

### 4.1.4 BASELINE METHODS

We compare our proposed methodology with the following four baseline methods in our experiments.

- *Random*: This method collects training samples by random exploration. We consider it to be an important baseline because of its simplicity and prevalence in a number of research works on self-supervised IL (Agrawal et al., 2016; Nair et al., 2017; Pathak et al., 2018).
- *Demo*: This method trains the imitator directly with expert demonstrations. It serves as the performance upper bound, as the training data is the same as the testing data for this method.
- *Curiosity*: This method trains a DRL agent via curiosity (Pathak et al., 2017; 2018) to collect training samples. Unlike the original implementation, we replace its DRL algorithm with PPO, as training should be done on a single thread for a fair comparison with the other baselines. This is alo an important baseline due to its effectiveness in Pathak et al. (2018).
- *Noise* (Plappert et al., 2018a): In this method, noise is injected to the parameter space of a DRL agent to encourage exploration (Plappert et al., 2018a). Please note that its exploratory behavior relies entirely on parameter space noise, instead of using any extrinsic reward. We include this method due to its superior performance and data efficiency in many DRL tasks.

## 4.2 PERFORMANCE COMPARISON IN ROBOTIC ARM MANIPULATION TASKS

We compare the performance of the proposed method and the baselines on the robotic arm manipulation tasks described in Section 4.1.1. As opposed to discrete control domains, these tasks are especially challenging, as the sample complexity grows in continuous control domains. Furthermore, the imitator may not have the complete picture of the environment dynamics, increasing its difficulty to learn an inverse dynamics model. In *FetchSlide*, for instance, the movement of the object on the slippery surface is affected by both friction and the force exerted by the gripper. It thus motivates us to investigate whether the proposed method can help overcome the challenge. In the subsequent paragraphs, we discuss the experimental results in both low- and high-dimensional observation spaces, and plot them in Figs. 2 and 3, respectively. All of the results are obtained by following the procedure described in Section 4.1.2. The shaded regions in Figs. 2 and 3 represent the confidence intervals.

**Low-dimensional observation spaces.** Fig. 2 plots the learning curves for all of the methods in low-dimensional observation spaces. In all of the tasks, our method yields superior or comparable performance to the baselines except for *Demo*, which is trained directly with expert demonstrations. In *FetchReach*, it can be seen that every method achieves a success rate of 1.0. This implies that it does not require a sophisticated exploration strategy to learn an inverse dynamics model in an environment where the dynamics is relatively simple. It should be noted that although all methods reach the same final success rate, ours learns significantly faster than *Demo*. In contrast, in *FetchPush*, our method is comparable to *Demo*, and demonstrates superior performance to the other baselines. Our method also learns drastically faster than all the other baselines, which confirms that the proposed strategy does improve the performance and efficiency of self-supervised IL. Our method is particularly effective in tasks that require an accurate inverse dynamics model. In *FetchPickAndPlace*, for example, our method surpasses all the other baselines. However, all methods including *Demo* fail to learn a successful inverse dynamics model in *FetchSlide*, which suggests that it is difficult to train an imitator when the outcome of an action is not completely dependent on the action itself. It is worth noting that *Curiosity* loses to *Random* in *FetchPush* and *FetchSlide*, and *Noise* performs even worse than these

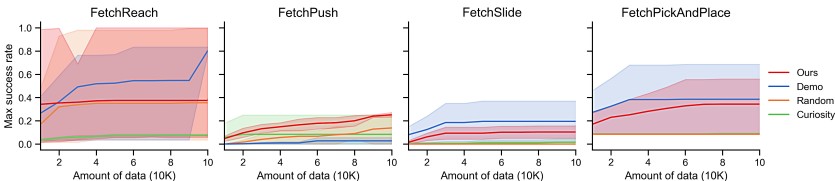

Figure 2: Performance comparison of robotic arm and hand tasks in low-dimensional observation spaces.

Figure 3: Performance comparison of robotic arm tasks in high-dimensional observation spaces.

two methods in all of the tasks. We therefore conclude that *Curiosity* is not suitable for continuous control tasks, and the parameter space noise strategy cannot be directly applied to self-supervised IL. In addition to the quantitative results presented above, we further discuss the empirical results qualitatively. Please refer our supplementary material for a description of the qualitative results.

**High-dimensional observation spaces.** Fig. 3 plots the learning curves of all methods in high-dimensional observation spaces. It can be seen that our method performs significantly better than the other baseline methods in most of the tasks, and is comparable to *Demo*. In *FetchPickAndPlace*, our method is the only one that learns a successful inverse dynamics model. Similar to the results in Fig. 2, *Curiosity* is no better than *Random* in high-dimensional observation spaces. Please note that we do not include *Noise* in Fig. 3 as it performs worse enough already in low-dimensional settings.

### 4.3 PERFORMANCE COMPARISON IN ROBOTIC HAND MANIPULATION TASK

Fig. 2 plots the learning curves for each of the methods considered. Please note that *Curiosity*, *Noise* and our method are pre-trained with 30K samples collected by random exploration, as we observe that these methods on their own suffer from large errors in an early stage during training, which prevents them from learning at all. After the first 30K samples, they are trained with data collected by their exploration strategy instead. From the results in Fig. 2, it can be seen that *Demo* easily stands out from the other methods as the best-performing model, surpassing them all by a considerable extent. Although our method is not as impressive as *Demo*, it significantly outperforms all of the other baseline methods, achieving a success rate of 0.4 while the others are still stuck at around 0.2.

The reason that the inverse dynamics models trained by the self-supervised data-collection strategies discussed in this paper (including ours and the other baselines) are not comparable to the *Demo* baseline in the *HandReach* task is primarily due to the high-dimensional action space. It is observed that the data collected by the self-supervised data-collection strategies only cover a very limited range of the state space in the *HandReach* environment. Therefore, the inverse dynamics models trained with these data only learn to imitate trivial poses, leading to the poor success rates presented in Fig. 2.

### 4.4 ROBUSTNESS TO ACTION SPACE PERTURBATION

We evaluate the performance of the imitator trained in an environment with action space perturbations to validate the robustness of our adversarial exploration strategy. In such an environment, every action taken by the DRL agent is perturbed by a Gaussian random noise, such that the training samples collected by the DRL agent are not inline with its actual intentions. Please note that we only inject noise during the training phase, as we aim to validate the robustness of the proposed data collection strategy. The scale of the injected noise is specified in the supplementary material. We report the performance change rates of various methods for different tasks in Table. 1. The performance change rate is defined as: $\frac{Pr_{perturb} - Pr_{orig}}{Pr_{orig}}$, where $Pr_{perturb}$ and $Pr_{orig}$ represent the highest success rates with and without action space perturbations, respectively. From Table. 1, it can be seen that our method retains the performance for most of the tasks, indicating that our method is robust to action space perturbations during the training phase. Please note that although *Curiosity* and *Noise* also achieve a change rate of 0% in *HandReach* and *FetchSlide*, they are not considered robust due to their poor performance in the original environment (Fig. 2). Another interesting observation is that our

|           | FetchReach | FetchPush | FetchSlide | FetchPickAndPlace | HandReach |
|-----------|------------|-----------|------------|-------------------|-----------|
| Random    | 0.00%      | -0.89%    | -23.21%    | -39.52%           | -32.32%   |
| Curiosity | 0.00%      | -45.48%   | -35.67%    | -18.61%           | 0.00%     |
| Noise     | 0.00%      | -90.00%   | **0.00%**  | **-12.03%**       | -40.00%   |
| Ours      | **0.00%**  | **1.64%** | -22.33%    | -23.17%           | **11.02%**|

Table 1: Comparison of performance change rate

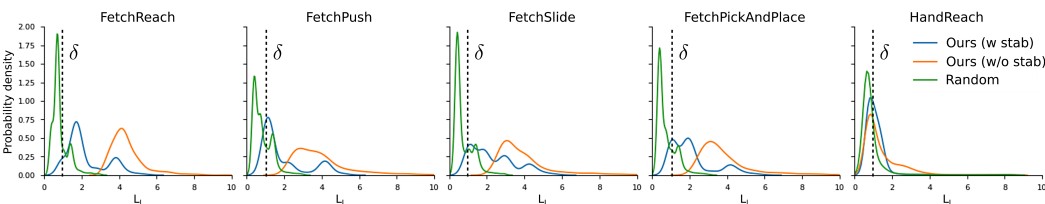

Figure 4: PDFs of $L_I$ in low-dimensional observation spaces for the first 2K training batches.

method even gains some performance from action space perturbations in *FetchPush* and *HandReach*, which we leave as one of our future directions. We thus conclude that our method is robust to action space perturbations during the training phase, making it a practical option in real-world settings.

## 4.5 ABLATIVE ANALYSIS

In this section, we provide a set of ablative analysis. We examine the effectiveness of our method by an investigation of the training loss distribution, the stabilization technique, and the influence of $\delta$. Please note that the value of $\delta$ is set to 1.5 by default, as described in our supplementary material.

**Training loss distribution.** Fig. 4 plots the probability density function (PDF) of $L_I$ (derived from Eq. (5)) by kernel density estimation (KDE) for the first 2K training batches during the training phase. The vertical axis corresponds to the probability density, while the horizontal axis represents the scale of $L_I$. The curves *Ours (w stab)* and *Ours (w/o stab)* represent the cases where the stabilization technique described in Section 3.3 is employed or not, respectively. We additionally plot the curve *Random* in Fig. 4 to highlight the effectiveness of our method. It can be observed that both *Ours (w stab)* and *Ours (w/o stab)* concentrate on notably higher loss values than *Random*. This observation implies that adversarial exploration strategy does explore hard samples for inverse dynamics model.

**Validation of the stabilization technique.** We validate the proposed stabilization technique in terms of the PDF of $L_I$ and the learning curve of the imitator, and plot the results in Figs. 4 and 5, respectively. From Fig. 4, it can be observed that the modes of *Ours (w stab)* are lower than those of *Ours (w/o stab)* in most cases, implying that the stabilization technique indeed motivates the DRL agents to favor those moderately hard samples. We also observe that for each of the five cases, the mode of *Ours (w stab)* is close to the value of $\delta$ (plotted in a dotted line), indicating that our reward structure presented in Eq. (7) does help to regulate $L_I$ (and thus $r_t$) to be around $\delta$. To further demonstrate the effectiveness of the stabilization technique, we compare the learning curves of *Ours (w stab)* and *Ours (w/o stab)* in Fig. 5. It is observed that for the initial 10K samples of the five cases, the success rates of *Ours (w/o stab)* are comparable to those of *Ours (w stab)*. However, their performance degrade drastically during the rest of the training phase. This observation confirms that the stabilization technique does contribute significantly to our adversarial exploration strategy.

Although most of the DRL works suggest that the rewards should be re-scaled or clipped within a range (e.g., from -1 to 1), the unbounded rewards do not introduce any issues during the training process of our experiments. The empirical rationale is that the rewards received by the DRL agent are regulated by Eq. (7) to be around $\delta$, as described in Section 4.5 and depicted in Fig. 4. Without the stabilization technique, however, the learning curves of the inverse dynamics model degrade drastically (as illustrated in Fig. 2), even if the reward clipping technique is applied.

**Influence of $\delta$.** Fig. 6 compares the learning curves of the imitator for different values of $\delta$. For instance, *Ours(0.1)* corresponds to $\delta = 0.1$. It is observed that for most of the tasks, the success rates drop when $\delta$ is set to an overly high or low value (e.g., 100.0 or 0.0), suggesting that a moderate value of $\delta$ is necessary for the stabilization technique. The value of $\delta$ can be adjusted dynamically by the adaptive scaling technique presented in Plappert et al. (2018a), which is left as our future direction.

From the analysis presented above, we conclude that the proposed adversarial exploration strategy is effective in collecting difficult training data for the imitator. The analysis also validates that our

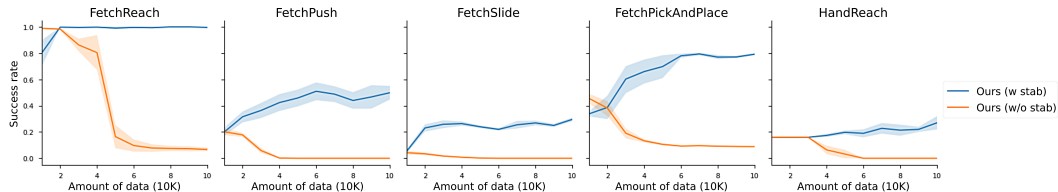

Figure 5: Learning curves w/ and w/o the stabilization technique in low-dimensional observation spaces.

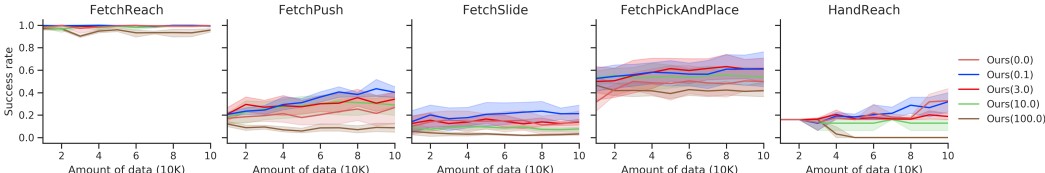

Figure 6: Performance comparison for different values of $\delta$.

stabilization technique indeed leads to superior performance, and is capable of guiding the DRL agent to collect moderately hard samples. This enables the imitator to pursue a stable learning curve.

## 5 CONCLUSION

In this paper, we presented an adversarial exploration strategy, which consists of a DRL agent and an inverse dynamics model competing with each other for self-supervised IL. The former is encouraged to adversarially collect difficult training data for the latter, such that the training efficiency of the latter is significantly enhanced. Experimental results demonstrated that our method substantially improved the data collection efficiency in multiple robotic arm and hand manipulation tasks, and boosted the performance of the inverse dynamics model in both low- and high-dimensional observation spaces. In addition, we validated that our method is generalizable to environments with high-dimensional action spaces. Moreover, we showed that our method is robust to action space perturbations. Finally, we provided a set of ablative analysis to validate the effectiveness for each of our design decisions.

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

# Supplementary Material

## S.1  QUALITATIVE ANALYSIS OF ROBOTIC ARM MANIPULATION TASKS

In addition to the quantitative results presented above, we further discuss the empirical results qualitatively. Through visualizing the training progress, we observe that our method initially acts like *Random*, but later focuses on interacting with the object in *FetchPush*, *FetchSlide*, and *FetchPickAnd-Place*. This phenomenon indicates that adversarial exploration strategy naturally gives rise to a curriculum that improves the learning efficiency, which resembles curriculum learning (Bengio et al., 2009). Another benefit that comes with the phenomenon is that data collection is biased towards interactions with the object. Therefore, the DRL agent concentrates on collecting interesting samples that has greater significance, rather than trivial ones. For instance, the agent prefers pushing the object to swinging the robotic arm. On the other hand, although *Curiosity* explores the environment very thoroughly in the beginning by stretching the arm into numerous different poses, it quickly overfits to one specific pose. This causes its forward dynamics model to keep maintaining a low error, making it less curious about the surroundings. Finally, we observe that the exploratory behavior of *Noise* does not change as frequently as ours, *Random*, and *Curiosity*. We believe that the method's success in the original paper (Plappert et al., 2018a) is largely due to extrinsic rewards. In the absence of extrinsic rewards, however, the method becomes less effective and unsuitable for data collection, especially in self-supervised IL.

## S.2  PROXIMAL POLICY OPTIMIZATION (PPO)

We employ PPO (Schulman et al., 2017) as the RL agent responsible for collecting training samples because of its ease of use and good performance. PPO computes an update at every timestep that minimizes the cost function while ensuring the deviation from the previous policy is relatively small. One of the two main variants of PPO is a clipped surrogate objective expressed as:

$$L^{CLIP}(\theta) = \mathbb{E}\left[\frac{\pi_\theta(a|s)}{\pi_{\theta_{old}}(a|s)}\hat{A}, \text{clip}(\frac{\pi_\theta(a|s)}{\pi_{\theta_{old}}(a|s)}, 1 - \epsilon, 1 + \epsilon)\hat{A})\right],$$

where $\hat{A}$ is the advantage estimate, and $\epsilon$ a hyperparameter. The clipped probability ratio is used to prevent large changes to the policy between updates. The other variant employs an adaptive penalty on KL divergence, given by:

$$L^{KLPEN}(\theta) = \mathbb{E}\left[\frac{\pi_\theta(a|s)}{\pi_{\theta_{old}}(a|s)}\hat{A} - \beta KL\left[\pi_{\theta_{old}}(\cdot|s), \pi_\theta(\cdot|s)\right]\right],$$

where $\beta$ is an adaptive coefficient adjusted according to the observed change in the KL divergence. In this work, we employ the former objective due to its better empirical performance.

## S.3  IMPLEMENTATION DETAILS OF INVERSE DYNAMICS MODEL

In the experiments, the inverse dynamics model $I(x_t, x_{t+1}|h_t, \theta_I)$ of all methods employs the same network architecture. For low-dimensional observation setting, we use 3 Fully-Connected (FC) layers with 256 hidden units followed by $tanh$ activation units. For high-dimensional observation setting, we use 3-layer Convolutional Neural Network (CNN) followed by $relu$ activation units. The CNNs are configured as (32, 8, 4), (64, 4, 2), and (64, 3, 1), with each element in the 3-tuple denoting the number of output features, width/height of the filter, and stride. The features extracted by stacked CNNs are then fed forward to a FC with 512 hidden units followed by $relu$ activation units.

## S.4  IMPLEMENTATION DETAILS OF ADVERSARIAL EXPLORATION STRATEGY

For both low- and high- dimensional observation settings, we use the architecture proposed in Schulman et al. (2017). During training, we periodically update the DRL agent with a batch of transitions as described in Algorithm. 1. We split the batch into several mini-batches, and update the RL agent with these mini-batches iteratively. The hyperparameters are listed in Table. 2 (our method).

## S.5  IMPLEMENTATION DETAILS OF *Curiosity*

Our baseline *Curiosity* is implemented based on the work (Pathak et al., 2018). The authors in Pathak et al. (2018) propose to employ a curiosity-driven RL agent (Pathak et al., 2017) to improve the

| Hyperparameter | Value |
|---|---|
| **Common** | |
| Batch size for inverse dynamic model update | 64 |
| Learning rate of inverse dynamic model | 1e-3 |
| Timestep per episode | 50 |
| Optimizer for inverse dynamic model | Adam |
| **Our method** | |
| Number of batch for update inverse dynamic model | 25 |
| Batch size for RL agent | 2050 |
| Mini-batch size for RL agent | 50 |
| Number of training iteration ($N_{iter}$) | 200 |
| Number of training episode per iteration ($N_{episode}$) | 10 |
| Horizon ($T$) of RL agent | 50 |
| Update period of RL agent | 2050 |
| Learning rate of RL agent | 1e-3 |
| Optimizer for RL agent | Adam |
| $\delta$ of stabilization | 1.5 |
| **Curiosity** | |
| Number of batch for update inverse dynamic model | 500 |
| Batch size for RL agent | 2050 |
| Mini-batch size for RL agent | 50 |
| Number of training iteration ($N_{iter}$) | 10 |
| Number of training episode per iteration ($N_{episode}$) | 200 |
| Horizon ($T$) of RL agent | 50 |
| Update period of RL agent | 2050 |
| Learning rate of RL agent | 1e-3 |
| Optimizer for RL agent | Adam |
| **Noise** | |
| Number of batch for update inverse dynamic model | 500 |
| The other hyperparameters | Same as Plappert et al. (2018a) |

Table 2: Hyperparameters settings.

efficiency of data collection. The curiosity-driven RL agent takes curiosity as intrinsic reward signal, where curiosity is formulated as the error in an agents ability to predict the consequence of its own actions. This can be defined as a forward dynamics model:

$$\hat{\phi}(x') = f(\phi(x), a; \theta_F), \tag{8}$$

where $\hat{\phi}(x')$ is the predicted feature encoding at the next timestep, $\phi(x)$ the feature vector at the current timestep, $a$ the action executed at the current timestep, and $\theta_F$ the parameters of the forward model $f$. The network parameters $\theta_F$ is optimized by minimizing the loss function $L_F$:

$$L_F\left(\phi(x), \hat{\phi}(x')\right) = \frac{1}{2}||\hat{\phi}(x') - \phi(x_{t+1})||_2^2. \tag{9}$$

For low- and high- dimensional observation settings, we use the architecture proposed in Schulman et al. (2017). The implementation of $\phi$ depends on the model architecture of the RL agent. For low-dimensional observation setting, we implement $\phi$ with the architecture of low-dimensional observation PPO. Note that $\phi$ does not share parameters with the RL agent in this case. For high-dimensional observation setting, we share the features extracted by the CNNs of the RL agent, then feed these features to $\phi$ which consists of a FC with 512 hidden units followed by $relu$ activation. The hyperparameters settings can be found in Table. 2(Curiosity).

## S.6 IMPLEMENTATION DETAILS OF *Noise*

We directly apply the same architecture in Plappert et al. (2018a) without any modification. Please refer to Plappert et al. (2018a) for more detail.

## S.7   IMPLEMENTATION DETAILS OF *Demo*

We collect 1000 episodes of expert demonstrations using the procedure defined in Sec. S8 for training *Demo*. Each episodes lasts 50 timesteps. The demonstration data is in the form of a 3-tuple $(x_t, a, x_{t+1})$, where $x_t$ is the current observation, $a_t$ the action, and $x_{t+1}$ the next observation. The pseudocode for training *Demo* is shown in Algorithm. S1 below. In each training iteration, we randomly sample 200 episodes, namely 10k transitions (line 4). The sampled data is then used to update the inverse dynamics model (line 5).

---

**Algorithm 2** *Demo*

---

1: Initialize $Z_{Demo}, \theta_I$
2: Set constants $N_{iter}$
3: **for** iter $i = 1$ to $N_{iter}$ **do**
4:     Sample 200 episodes of demonstration from $Z_{Demo}$ as $B$
5:     Update $\theta_I$ with the gradient calculated from the samples in $B$ (according to Eq. 6)
6: **end**

---

## S.8   CONFIGURATION OF ENVIRONMENTS

We briefly explain each configuration of the environment below. For detailed description, please refer to Plappert et al. (2018b).

- *FetchReach:* Control the gripper to reach a goal position in 3D space. The imitator can fully comprehend the environment dynamics.
- *FetchPush:* Control the Fetch robot to push the object to a target position. The imitator cannot fully comprehend the environment as the movement of the gripper may not affect the object.
- *FetchPickAndPlace:* Control the gripper to grasp and lift the object to a goal position. In addition to the imitator not having the complete picture of the environment dynamics, this task requires a more accurate inverse dynamics model.
- *FetchSlide:* Control the robot to slide the object to a goal position. The task requires an even more accurate inverse dynamics model, as the object's movement on the slippery surface is hard to predict.
- *HandReach:* Control the Shadow Dextrous Hand to reach a goal hand pose. The task is especially challenging due to high-dimensional action spaces.

## S.9   SETUP OF EXPERT DEMONSTRATION

We employ Deep Deterministic Policy Gradient combined with Hindsight Experience Replay (DDPG-HER) (Andrychowicz et al., 2017) as the expert agent. For training and evaluation, we run the expert to collect transitions for 1000 and 500 episodes, respectively. To prevent the imitator from succeeding in the task without taking any action, we only collect successful and non-trivial episodes generated by the expert agent. Non-trivial episodes are filtered out based on the following task-specific schemes:

- *FetchReach:* An episode is considered trivial if the distance between the goal position and the initial position is smaller than 0.2.
- *FetchPush:* An episode is determined trivial if the distance between the goal position and the object position is smaller than 0.2.
- *FetchSlide:* An episode is considered trivial if the distance between the goal position and the object position is smaller than 0.1.
- *FetchPickAndPlace:* The episode is considered trivial if the distance between the goal position and the object position is smaller than 0.2.
- *HandReach:* We do not filter out trivial episodes as this task is too difficult for most of the methods.

## S.10   ANALYSIS OF THE NUMBER OF EXPERT DEMONSTRATIONS

Fig. 7 illustrates the performance of *Demo* with different number of expert demonstrations. *Demo(100)*, *Demo(1,000)*, and *Demo(10,000)* correspond to the *Demo* baselines with 100, 1,000, and 10,000 episodes of demonstrations, respectively. It is observed that their performance are comparable for most of the tasks except *FetchReach*. In *FetchReach*, the performance of *Demo(100)* is significantly worse than the other two cases. A possible explanation is that preparing a sufficiently diverse

set of demonstrations in *FetchReach* is relatively difficult with only 100 episodes of demonstrations. A huge performance gap is observed when the number of episodes is increased to 1,000. Consequently, *Demo(1,000)* is selected as our *Demo* baseline for the presentation of the experimental results in Section 4. Another advantage is that *Demo(1,000)* demands less memory than *Demo(10,000)*.

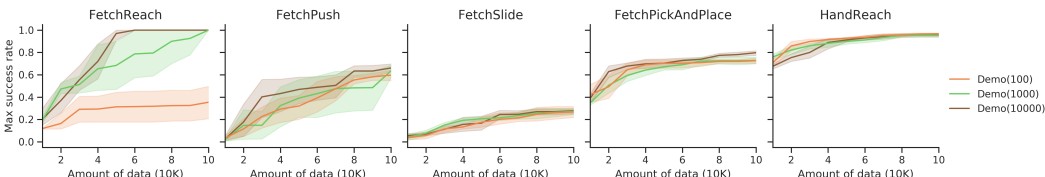

Figure 7: Comparison of different number of expert demonstrations in low-dimensional observation spaces.

## S.11 SETUP OF NOISY ACTION

To test the robustness of our method to noisy actions, we add noise to the actions in the training stage. Let $\hat{a}_t$ denote the predicted action by the imitator. The actual noisy action to be executed by the robot is defined as:

$$\hat{a}_t := \hat{a}_t + \mathcal{N}(0, \sigma),$$

where $\sigma$ is set as $0.01$. Note that $\hat{a}_t$ will be clipped in the range defined by each environment.

## S.12 VISUALIZATION OF STABILIZATION TECHNIQUE

In this section, we visualize the effects of our stabilization technique with a list of rewards $r$ in Fig. 8. The rows of *Before* and *After* represent the rewards before and after reward shaping, respectively. The bar on the right-hand side indicates the scale of the reward. It can be observed in Fig. 8 that after reward shaping, the rewards are transformed to the negative distance to the specified $\delta$ (i.e., 2.5 in this figure). As a result, our stabilization technique is able to encourage the DRL agent to pursue rewards close to $\delta$, where higher rewards can be received.

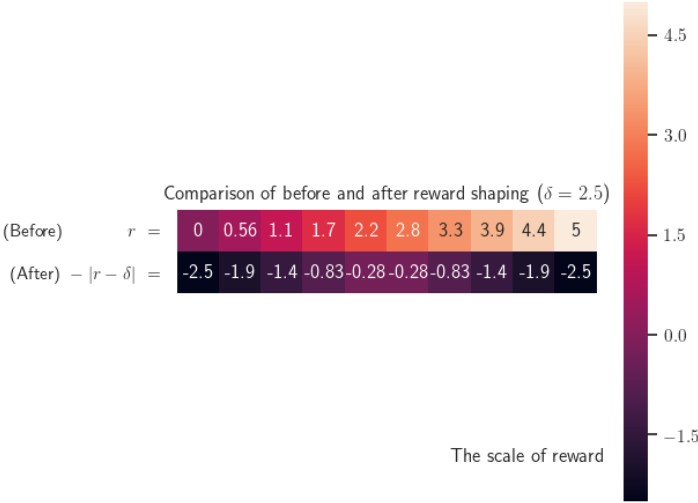

Figure 8: Visualization of the stabilization technique.

