# OpenReview forum: "Adversarial Exploration Strategy for Self-Supervised Imitation Learning"
_ICLR.cc/2019/Conference_

### Official Review · AnonReviewer3 · 2018-11-02
**Cool idea, but there are issues in evaluation and results are weak overall**

**Rating:** 5
**Confidence:** 3

**Review:**

The paper proposes a novel exploration strategy for self-supervised imitation learning. An inverse dynamics model is trained on the trajectories collected from a RL-trained policy. The policy is rewarded for generating trajectories on which the inverse dynamics model (IDM) currently works poorly, i.e. on which IDM predicts actions that are far (in terms of mean square error) from the actions performed by the policy. This adversarial training is performed in purely self-supervised way. The evaluation is performed by one-shot imitation of an expert trajectory using the IDM: the action is predicted from the current state of the environment and the next state in the expert’s trajectory. Experimental evaluation shows that the proposed method is superior to baseline exploration strategies for self-supervised imitation learning, including random and curiosity-based exploration.

Overall, I find the idea quite appealing. I am not an expert in the domain and can not make comments on the novelty of the approach. I found the writing mostly clear, except for the following issues:
- the introduction has not made it crystal clear that the considered paradigm is different from e.g. DAGGER and GAIL in that expert demonstrations are used at the inference time. A much wider audience is familiar with the former methods, and this distinction should have be explained more clearly.
- Section 4.2.: “As opposed to discrete control domains, these tasks are especially challenging, as the sample complexity grows in continuous control domains.” - this sentence did not make sense to me. It basically says continuous control is challenging because it is challenging.
- I did not understand the stabilization approach. How exactly Equation (7) forces the policy to produce “not too hard” training examples for IDM? Figure 4 shows that it is on the opposite examples with small L_I that are avoided by using \delta > 0.
- Table 1 - it is a bit counterintuitive that negative numbers are better than positive numbers here. Perhaps instead of policy’s deterioration you could report the relative change, negative when the performance goes down and positive otherwise?

I do have concerns regarding the experimental evaluation:
- the “Demos” baseline approach should be explained in the main text! In Appendix S.7 I see that 1000 human demonstrations were used for training. Why 1000, and not 100 and not 10000?  How would the results change? This needs to be discussed. Without discussing this it is really unclear how the proposed method can outperform “Demos”, which it does pretty often.
- it is commendable that 20 repetitions of each experiment were performed, but I am not sure if it is ever explained in the paper what exactly the upper and lower boundaries in the figures mean. Is it the standard deviation? A confidence interval? Can you comment on the variance of the proposed approach, which seems to be very high, especially when I am looking at high-dimensional fetch-reach results?
- the results of “HandReach” experiments, where the proposed method works much worse than “Demos” are not discussed in the text at all
- overall, there is no example of the proposed method making a difference between a “working” and “non-working” system, compared to “Curiosity” and “Random”. I am wondering if improvements from 40% to 60% in such cases are really important. In 7 out of 9 plots the performance of the proposed method is less than 80% - not very impressive. "Demos" baseline doesn't perform much better, but what would happen with 10000 demonstrations?
- there is no comparison to behavioral cloning, GAIL, IRL. Would these methods perform better than learning IDM like "Demos" does?

I think that currently the paper is slightly below the threshold, due to evaluation issues discussed above and overall low performance of the proposed algorithm. I am willing to reconsider my decision if these issues are addressed.

---

> ### Author Response · Authors · 2018-11-16
> **Response to reviewer 3 (Part 1/3)**
>
> Here is the PDF version of our responses: https://www.dropbox.com/s/mxhetdyy7m4nkp6/Response_3_ICLR_2019.pdf?dl=0 (anonymous link)
>
> The authors appreciate the reviewer’s the time and efforts for reviewing this paper, and would like to respond to the questions in the following paragraphs.
>
> Q1: The introduction has not made it crystal clear that the considered paradigm is different from e.g. DAGGER and GAIL in that expert demonstrations are used at the inference time. A much wider audience is familiar with the former methods, and this distinction should have be explained more clearly.
>
> Response: The authors appreciate the thoughtful feedbacks from the reviewer, and would like to bring to the reviewer’s kind attention that in Section 1, the following sentence for describing this distinction had been included in the original manuscript:
>
> “Self-supervised IL allows an imitator to collect training data by itself instead of using predefined extrinsic reward functions or expert supervision during training. It only needs demonstration during inference, drastically decreasing the time and effort required from human experts.”
>
> Q2: Section 4.2.: “As opposed to discrete control domains, these tasks are especially challenging, as the sample complexity grows in continuous control domains.” - this sentence did not make sense to me. It basically says continuous control is challenging because it is challenging.
>
> Response:  Thanks for sharing your thoughts with us.  We would love to clarify the meaning of this sentence as follows.
>
> It is challenging to train inverse dynamics models in continuous control domains because of its requirement of diverse sample data during the training phase.  The diverse training data should cover various state transitions (i.e., (x_t, x_{t+1})) as well as their corresponding actions (i.e., a_t).  To this end, an explorer (i.e., the self-supervised data-collection DRL agent) is required to visit a wide range of states and extensively attempt different actions.    Such a requirement can be easily achieved in discrete control domains, as their action spaces are not as large as those of the continuous domain counterparts, allowing an explorer to quickly travel through most of possible state transitions.  In contrast, due to the enormous number of potential actions, the sample complexity (https://en.wikipedia.org/wiki/Sample_complexity) of continuous control domains is significantly higher than that of discrete ones.  This is the reason we called it challenging in the original manuscript.
>
> We sincerely hope that we have adequately addressed your concerns.
>
> Q3: I did not understand the stabilization approach. How exactly Eq. (7) forces the policy to produce “not too hard” training examples for IDM? Fig. 4 shows that it is on the opposite examples with small L_I that are avoided by using δ > 0.
>
> Response:  The main objective of Eq. (7) is to shape the rewards (i.e., the losses of the inverse dynamics model) to be the negative L1-distance to δ.  In other words, the closer the original unshaped reward is to δ, the higher the shaped reward is.  As a result, the stabilization technique presented in Eq. (7) encourages the RL agent to collect “moderately difficult” samples that regulates the losses of the inverse dynamics model to be around δ.
>
> Q4: Table 1 - it is a bit counterintuitive that negative numbers are better than positive numbers here. Perhaps instead of policy’s deterioration you could report the relative change, negative when the performance goes down and positive otherwise?
>
> Response:  The authors appreciate the thoughtful feedbacks from the reviewer, and have rephrased the term as “performance change rate” and made the other necessary revisions in Section 4.4 according to the suggestions in the updated manuscript.
>
> Q5: The “Demos” baseline approach should be explained in the main text! In Appendix S.7 I see that 1000 human demonstrations were used for training. Why 1000, and not 100 and not 10000?  How would the results change? This needs to be discussed. Without discussing this it is really unclear how the proposed method can outperform “Demos”, which it does pretty often.
>
> Response:  We fully agree with your comments regarding the number of demonstrations.  To address your concerns, we have incorporated an additional figure illustrating the learning curves of the Demo baseline with various number of demonstrations in the supplementary material.  According to Fig. 7, we do not observe any significant difference in performance when the number of demonstrations is set to 100, 1,000, and 10,000.  This is the reason why we used 1,000 demonstrations for training this baseline method in our experiments.

---

> > ### Author Response · Authors · 2018-11-16
> > **Response to reviewer 3 (Part 2/3)**
> >
> > Q6: It is commendable that 20 repetitions of each experiment were performed, but I am not sure if it is ever explained in the paper what exactly the upper and lower boundaries in the figures mean. Is it the standard deviation? A confidence interval?
> >
> > Response:  We would like to thank the reviewer for pointing out this issue, and sincerely apologize for the confusion caused.  In Figs. 2, 3, 5, and 6, the shaded regions of the curves correspond to their confidence intervals.  We have enhanced the manuscript with additional descriptions in the revised version.
> >
> > Q7: Can you comment on the variance of the proposed approach, which seems to be very high, especially when I am looking at high-dimensional fetch-reach results?
> >
> > Response:  The variance in the learning curves is mainly caused by the complexity of the high-dimensional observation spaces.  Contemporary DRL techniques also suffer from the same problems when training with raw images (i.e., high-dimensional observation spaces) [1].  The high variance in the learning curves can be alleviated by enhancing the model architecture or the training algorithm.  Please note that this issue also occurs in the learning curves of the other baseline methods.  Discussion of model architectures and specific training techniques for high-dimensional observation spaces, however, is beyond the scope of this paper.
> >
> > Q8: The results of “HandReach” experiments, where the proposed method works much worse than “Demos” are not discussed in the text at all
> >
> > Response:  The reason that the inverse dynamics models trained by the self-supervised data-collection strategies discussed in this paper (including ours and the other baselines) are not comparable to the Demo baseline in the HandReach task is primarily due to the high-dimensional action space.  According to our experiments, it is observed that the data collected by the self-supervised data-collection strategies only cover a very limited range of the state space in the HandReach environment.  Therefore, the inverse dynamics model trained with these data only learn to imitate trivial poses, leading to the poor success rates presented in Fig. 2.  We would like to thank the reviewer for the suggestion, and have enhanced the manuscript with the above explanation in the revised version.
> >
> > Q9: Overall, there is no example of the proposed method making a difference between a “working” and “non-working” system, compared to “Curiosity” and “Random”. I am wondering if improvements from 40% to 60% in such cases are really important. In 7 out of 9 plots the performance of the proposed method is less than 80% - not very impressive.
> >
> > Response:  We understand the reviewer’s concerns.  However, we do have some reservations about the perspective of the comments, and would like to discuss our point of views with the reviewer in two different aspects.  First, the main scope of this work is self-supervised data-collection strategies for training inverse dynamics models.  The baseline methods selected for comparison are mostly published in this year.  As demonstrated in Section 4, our method outperforms them significantly for most of the tasks in both low- and high-dimensional observation spaces, as well as high-dimensional action space.  Second, to the best of our knowledge, almost none of the previous works in this domain has investigated adversarial exploration strategies for training inverse dynamics models.  Our work is the first proof of concept to demonstrate that the proposed adversarial strategy does improve the training efficiency and the performance of inverse dynamics models.  We hope that the reviewer could kindly correct us if we are wrong, and consider the points discussed above.

---

> > > ### Author Response · Authors · 2018-11-16
> > > **Response to reviewer 3 (Part 3/3)**
> > >
> > > Q10:  "Demos" baseline doesn't perform much better, but what would happen with 10,000 demonstrations?
> > >
> > > Response:  To address your concerns, we have incorporated an additional figure illustrating the learning curves of the Demo baseline with various number of demonstrations in in Section S10 of the supplementary material.  According to Fig. 7, we do not observe any significant difference in performance when the number of demonstrations is set to 100, 1,000, and 10,000. Hence, we use 1,000 demonstrations for training the baselines as it costs less memory than the others and has comparable performance.  We have also revised Section S10 in the supplementary material with the above explanation.
> > >
> > > Q11: There is no comparison to behavioral cloning, GAIL, IRL. Would these methods perform better than learning IDM like "Demos" does?
> > >
> > > Response:  We would like to bring to the reviewer’s kind attention that the primary focus of this work is self-supervised imitation learning (IL), rather than traditional IL (e.g., GAIL and IRL).  The formulation of these two branches are fundamentally different from each other.  Self-supervised IL takes demonstrations in the testing phase only.  As a result, it allows the tasks to be altered online by changing the contents (i.e., trajectories) of the demonstrations.  On the other hand, traditional IL uses demonstrations in the training phase, and does not allow online changes to the tasks in the testing phase.  Therefore, we consider that these two branches are not supposed to be compared due to their distinct problem formulations.
> > >
> > > [1] M. Fortunato et al., "Noisy networks for exploration," in Proc. Int. Conf. Learning Representations (ICLR), Apr.-May, 2018.

---

> > > > ### Comment · AnonReviewer3 · 2018-11-25
> > > > **Feedback from Reviewer 3**
> > > >
> > > > Dear Authors,
> > > >
> > > > Thank you for your clarifications, they were quite helpful. I think you should consider editing the spots of the paper that caused confusion, in particular, you could explain a gist of how self-supervised IL works as early as in the intro (1 sentence), and elaborate more on the sample complexity differences between discrete and continuous action spaces.
> > > >
> > > > In Eq. 7 you are abusing the notation, by using r_t both on the right-hand side and the left-hand side. This is confusing. You could use e.g. \hat{r}_t on the left-hand side. Also, this transformation is not really reward shaping in sense of (Ng et al, 1999) (https://goo.gl/t68wpH), which is the sense in which most of the community understand this term. Lastly, your plots in Figure 4 seem to be showing that the transformation performed in Eq. 7 decreases first and foremost the frequency of easy examples, and not the difficult ones (I’m comparing w/o stab and w. stab). I don’t think you discuss this discrepancy in the paper or in your response.
> > > >
> > > > Thank you for commenting on the number of demonstrations. I am still not fully satisfied though: my understanding is that given enough examples, behavioural cloning should eventually perform perfectly, which is not the case, according to Fig. 7. Do you train for not long enough? Not having a clear baseline makes it hard to put your results in context and understand how useful they actually are.
> > > >
> > > > I guess the main disagreement between myself and the authors of the paper have is the question of how such work should be positioned and evaluated. The authors suggest that their method should only be compared only to other methods from the narrow subfield of learning general purpose inverse models without any supervision and then using them for what effectively is one-shot imitation learning (at least that’s my understanding). They argue that any statistically significant improvement upon baselines from this narrow subfield is sufficient for publication. With all due respect to the hard work that the authors have conducted, I am inclined to disagree. My opinion is that these methods should be considered in a broader context of imitation learning in general. Yes, methods such as behavioral cloning, GAIL and IRL are less generic than the proposed one and use more expert data, but on the other hand they come with a clear guarantee that the more data you collect the better your performance is, and for the time being are probably superior to fully unsupervised methods. I believe that research such as this paper should make it clear how much of the gap between fully unsupervised imitation learning and conventional imitation learning is bridged by the proposed method. I believe conventional imitation learning baselines should be established with better diligence than what this work does. The fact that on the majority of tasks “Demos” baseline is so far from 100% leaves me puzzled. Was it not enough training iterations? Would GAIL perform a lot of better?
> > > >
> > > > You may want to consider studying in your future research how helpful your method is when combined with behavioral cloning or GAIL. That could be a compelling argument. You may try and explain better why all methods (including “Demos”) perform so poorly on the majority of tasks. But as is, I am not convinced that the paper is ready to be presented. This is my opinion, and the AC will have the last word, of course.

---

> > > > > ### Author Response · Authors · 2018-12-05
> > > > > **Response to reviewer 3 (Part 1/4)**
> > > > >
> > > > > Here is the PDF version of our response: https://www.dropbox.com/s/fyzs56wbjyb6ykp/20181215%20Replies%20to%20Reviewer%203.pdf?dl=0 (anonymous link)
> > > > >
> > > > > The authors appreciate the reviewer’s the time and efforts for reviewing this paper, and would like to respond to the questions in the following paragraphs.
> > > > >
> > > > > Comment: In Eq. 7 you are abusing the notation, by using r_t both on the right-hand side and the left-hand side.  This is confusing.  You could use e.g. \hat{r}_t on the left-hand side.
> > > > >
> > > > > Response:
> > > > > We would like to thank the reviewer for the suggestion.  We will be glad to enhance the readability of Eq. (7) in the final version of the manuscript.
> > > > >
> > > > > Comment: Also, this transformation is not really reward shaping in sense of (Ng et al, 1999) (https://goo.gl/t68wpH), which is the sense in which most of the community understand this term.
> > > > >
> > > > > Response:
> > > > > The authors would like to thank the reviewer for sharing his/her thoughts with us.  However, we do have some concerns about the perspective of the comments, and do not fully agree with the reviewer’s point of view.  We would like to discuss our point of view with the reviewer in two different perspectives.
> > > > >
> > > > > First, we would like to bring to the reviewer’s attention that the stabilization technique presented in Eq. (7) does meet the definition of “reward shaping” in [2].  The 3rd footnote of [2] states that the concept of reward shaping is considered more general, and is not restricted to the additive form.  In their definition, the shaped reward function R’(s, a, s’) can be defined as F(r, s, a, s’), where F is an arbitrary function.  As a result, Eq. (7) in our paper does not violate the meaning of “reward shaping” defined in [2].
> > > > >
> > > > > Second, there have been several recent research works [3, 4] in deep reinforcement learning (DRL) adopting the term “reward shaping” for various objectives, with their shaping function defined by following the general from in [2].  Therefore, we consider that Eq. (7) in our paper conforms to the understanding of “reward shaping” accepted by the general community.
> > > > >
> > > > > We sincerely hope that we have adequately addressed the reviewer’s concerns.
> > > > >
> > > > > [2] A. Ng, D. Harada, and S. J. Rusell, “Policy invariance under reward transformations: Theory and application to reward shaping,” in Proc. Int. Conf. Machine Learning (ICML), Jun. 1999, pp. 278-287.
> > > > > [3] M. Andrychowicz et al., "Hindsight experience replay," in Proc. the 31st Conf. Neural Information Processing Systems (NIPS), Dec. 2017.
> > > > > [4] V. Pong, S. Gu, M. Dalal, and S. Levine "Temporal difference models: Model-free deep RL for model-based control," in Proc. Int. Conf. Learning Representations (ICLR), Apr.-May, 2018.
> > > > >
> > > > > Comment: Lastly, your plots in Figure 4 seem to be showing that the transformation performed in Eq. 7 decreases first and foremost the frequency of easy examples, and not the difficult ones (I’m comparing w/o stab and w. stab). I don’t think you discuss this discrepancy in the paper or in your response.
> > > > >
> > > > >
> > > > > Response:
> > > > > We are afraid that there seems to be some misunderstanding.  In Section 4.5, we discussed that for each of the five cases, the mode of Ours (w stab) is close to the value of δ (plotted in a dotted line), indicating that our reward structure presented in Eq. (7) does help to regulate L_I (and thus r_t) to be around δ.  In other words, the DRL agent is not motivated to collect easy samples (L_I < δ) or hard samples (L_I > δ).  It is only encouraged to collect moderately hard samples (L_I ≈ δ) to to train the inverse dynamics model.  We illustrate the effect of Eq. (7) by the following figure.
> > > > >
> > > > > https://www.dropbox.com/s/mhivrt9a1eoxhyd/fig_rebuttal_reward_shaping.png?dl=0
> > > > >
> > > > >
> > > > > We would also like to bring to the reviewer’s attention that in the revised manuscript we posted on OpenReview last time, we have included the above figure in the manuscript as Fig. 8 along with  an additional Section S.12 to visualize how rewards are shaped.
> > > > >
> > > > > In addition, in Section 4.5 of the original manuscript we have discussed that from Fig. 4, it can be observed that the modes of Ours (w stab) are lower than those of Ours (w/o stab) in most cases, implying that the stabilization technique does motivate the DRL agents to favor those moderately hard samples.
> > > > >
> > > > > We sincerely hope that we have adequately addressed your concerns.

---

> > > > > > ### Author Response · Authors · 2018-12-05
> > > > > > **Response to reviewer 3 (Part 2/4)**
> > > > > >
> > > > > > Comment:  Thank you for commenting on the number of demonstrations. I am still not fully satisfied though: my understanding is that given enough examples, behavioural cloning should eventually perform perfectly, which is not the case, according to Fig. 7.  Do you train for not long enough? Not having a clear baseline makes it hard to put your results in context and understand how useful they actually are.
> > > > > >
> > > > > > Response:
> > > > > > We would like to bring to the reviewer’s kind attention that the “Demo” baseline is different from behavior cloning.  For the “Demo” baseline presented in this paper, the inverse dynamics model is trained with the data obtained from expert demonstrations.  Behavior cloning (BC), on the other hand, does not train an inverse dynamics model.  Instead, BC requires a model to learn a policy directly from expert demonstrations.  We illustrate the difference between self-supervised imitation learning (IL) and traditional IL (e.g., BC, GAIL, etc) as the following:
> > > > > >
> > > > > > Traditional IL:
> > > > > > https://www.dropbox.com/s/bzebc4fw907x61s/Traditional_IL.jpeg?dl=0
> > > > > >
> > > > > > Self-Supervised IL:
> > > > > > https://www.dropbox.com/s/g6bycuzefa9u4k3/Self-Supervised_IL.jpeg?dl=0
> > > > > >
> > > > > > The reason why the performance of the “Demo” baseline is not perfectly 100% is attributable to the model architecture, which limits the final performance of the imitator.  Different model architectures do lead to different success rates of the self-supervised IL models.  As the main contribution of this paper is the adversarial exploration strategy for self-supervised imitation learning, we fixed the model architecture for all of our experiments.  Discussion of the most effective model architecture for IL is not within the scope of discussion of this paper.
> > > > > >
> > > > > > Fig. 7 illustrates the performance of Demo with different number of expert demonstrations.  Demo(100), Demo(1,000), and Demo(10,000) correspond to the Demo baselines with 100, 1,000, and 10,000 episodes of demonstrations, respectively.  It is observed that the three curves in Fig. 7 saturate to similar performance for most of the tasks.  This experiment can therefore serve as an evidence that the performance of the Demo baseline is not fully determined by the number of iterations.

---

> > > > > > > ### Author Response · Authors · 2018-12-05
> > > > > > > **Response to reviewer 3 (Part 3/4)**
> > > > > > >
> > > > > > > Comment:  I guess the main disagreement between myself and the authors of the paper have is the question of how such work should be positioned and evaluated.  The authors suggest that their method should only be compared only to other methods from the narrow subfield of learning general purpose inverse models without any supervision and then using them for what effectively is one-shot imitation learning (at least that’s my understanding).  They argue that any statistically significant improvement upon baselines from this narrow subfield is sufficient for publication.  With all due respect to the hard work that the authors have conducted, I am inclined to disagree.
> > > > > > >
> > > > > > > My opinion is that these methods should be considered in a broader context of imitation learning in general.  Yes, methods such as behavioral cloning, GAIL and IRL are less generic than the proposed one and use more expert data, but on the other hand they come with a clear guarantee that the more data you collect the better your performance is, and for the time being are probably superior to fully unsupervised methods.  I believe that research such as this paper should make it clear how much of the gap between fully unsupervised imitation learning and conventional imitation learning is bridged by the proposed method.  I believe conventional imitation learning baselines should be established with better diligence than what this work does.
> > > > > > >
> > > > > > > Response:
> > > > > > > We respectfully disagree with the reviewer’s opinion, as self-supervised imitation learning (IL) and traditional IL belong to two different branches of research directions.  In addition, the primary target of this paper is to enhance the training efficiency of self-supervised IL, rather than proposing a method to compete with existing IL methods.  We illustrate the difference between self-supervised imitation learning (IL) and traditional IL (e.g., BC, GAIL, etc) as the following:
> > > > > > >
> > > > > > > Traditional IL:
> > > > > > > https://www.dropbox.com/s/bzebc4fw907x61s/Traditional_IL.jpeg?dl=0
> > > > > > >
> > > > > > > Self-Supervised IL:
> > > > > > > https://www.dropbox.com/s/g6bycuzefa9u4k3/Self-Supervised_IL.jpeg?dl=0
> > > > > > >
> > > > > > > We summarize our perspectives in the following.
> > > > > > >
> > > > > > > First, as described in Section 1, self-supervised IL allows an imitator to collect training data by itself instead of using pre-defined extrinsic reward functions or expert supervision during training.  It only needs demonstration during inference, drastically decreasing the time and effort required from human experts.  In complex environments such as Mujoco [5], Roboschool [6], OpenAI robotic arm and hand task [7], and real robotic tasks, it is extremely difficult to collect sufficient expert demonstrations in reasonable amount of time.  Traditional methods of data collection are usually inefficient and time-consuming.  Inefficient data collection results in poor exploration, giving rise to a degradation in robustness to varying environmental conditions (e.g., noise in motor control) and generalizability to difficult tasks.  The proposed method enables an IL model to learn from the data prepared by the DRL agent, significantly removing the need of human intervention.
> > > > > > >
> > > > > > > The second difference is that during the training phase, self-supervised IL motivates an an inverse dynamics model based imitator to learn by interacting with the environment, while traditional IL requires a DRL agent based imitator to learn from expert demonstrations.
> > > > > > >
> > > > > > > The third difference is that during the evaluation phase, self-supervised IL requires an imitator to infer the transitional action (a_t) between the current observation (x_t) and the given observation (\hat{x}_{t+1}) from the expert’s demonstration, while traditional IL does not use any expert demonstration during the evaluation phase.  The imitator trained by traditional IL simply follows the policy learned during the training phase, and is unable to adapt to tasks not included in the training data.
> > > > > > >
> > > > > > > Finally, in terms of objectives, self-supervised IL aims at training an effective inverse dynamics model, and execute it in the evaluation phase by following an expert’s observations online.  On the other hand, traditional IL methods focus on learning a policy and perform predefined tasks only.
> > > > > > >
> > > > > > > In summary, self-supervised IL differs from traditional IL in several aspect, including their methods of training data preparation, training and evaluation procedures, as well as their objectives.  Hence, we consider that these two methods are not directly comparable because of the above reasons.
> > > > > > >
> > > > > > > [5] E. Todorov, T. Erez, and Y. Tassa, “Mujoco: A physics engine for model-based control view,” in Proc. IEEE/RSJ Int. Conf. Intelligent Robots and Systems (IROS), pp. 5026-5033, Oct. 2012.
> > > > > > > [6] Roboschool, available at https://blog.openai.com/roboschool/.
> > > > > > > [7] Ingredients for Robotics Research environments, available at https://blog.openai.com/ingredients-for-robotics-research/.

---

> > > > > > > > ### Author Response · Authors · 2018-12-05
> > > > > > > > **Response to reviewer 3 (Part 4/4)**
> > > > > > > >
> > > > > > > > Comment:  The fact that on the majority of tasks “Demos” baseline is so far from 100% leaves me puzzled.  Was it not enough training iterations?  Would GAIL perform a lot of better?
> > > > > > > >
> > > > > > > > Response:
> > > > > > > > We would like to address the reviewer’s questions in three aspects.
> > > > > > > >
> > > > > > > > First, we would like to bring to the reviewer’s kind attention that more iterations do not necessarily resulting to a higher success rate for the “Demo” baseline.  In the responses and the revised manuscript we posted on OpenReview last time, we have included an additional experiment in Section S.10 for comparing the performance of the Demo baseline trained with different number of iterations.  Fig. 7 illustrates the performance of Demo with different number of expert demonstrations.  Demo(100), Demo(1,000), and Demo(10,000) correspond to the Demo baselines with 100, 1,000, and 10,000 episodes of demonstrations, respectively.  It is observed that the three curves in Fig. 7 saturate to similar performance for most of the tasks.  This experiment can therefore serve as an evidence that the performance of the Demo baseline is not fully determined by the number of iterations.
> > > > > > > >
> > > > > > > > The gap between the real success rate and 100% for the Demo baseline is attributable to the model architecture, which limits the final performance of the imitator.  Different model architectures do lead to different success rates of the self-supervised IL models.  However, as the main contribution of this paper is an adversarial exploration strategy for self-supervised imitation learning, we fixed the model architecture for all of our experiments.  Discussion of the most effective model architecture for IL is not within the scope of discussion of this paper.
> > > > > > > >
> > > > > > > > Finally, we hope the reviewer could understand that GAIL and self-supervised IL are different in their problem formulations.  GAIL trains the imitator with expert demonstrations during the training phase, and uses the learned policy to perform predefined tasks in the evaluation phase without any additional demonstrations.  On the other hand, self-supervised IL aims at training an effective inverse dynamics model, and execute it in the evaluation phase by following an expert’s observations online.  GAIL is unable to perform this kind of tasks if the corresponding expert demonstrations are not provided during the training phase.  As a result, these two streams of methods are not comparable because of their distinct problem formulations.

---

### Official Review · AnonReviewer1 · 2018-11-02
**interesting and potentially useful paper**

**Rating:** 5
**Confidence:** 3

**Review:**

The paper proposes an exploration strategy for deep reinforcement learning agent in continuous action spaces. The core of the method is to train an inverse local model (a model that predicts the action that was taken from a pair of states) and its errors as an exploration bonus for a policy gradient agent. The intuition is that its a good self-regulating strategy similar to curiosity that leads the agents towards states that are less known by the inverse model. Seeing these states improves the . There are experiments run on the OpenAI gym comparing to other models of curiosity. The paper is well written and clear for the most part.

pros:
- the paper seems novel and results are promising
- easy to implement
cons:
- seems unstable and not clear how it would scale in a large state space where most states are going to be very difficult to learn about in the beginning like a humanoid body.
- only accounts for the immediately controllable aspects of the environment which doesn't seem to be the hard part. Understanding the rest of the environment and its relationship to the controllable part of the state seems beyond the scope of this model. Nonetheless I can imagine it helping with initial random motions.
- from (6) the bonus seems to be unbounded and (7) doesn't seem to fix that. Is that not an issue in general ? Any intuition about that ?

---

> ### Author Response · Authors · 2018-11-16
> **Response to reviewer 1 (Part 1/2)**
>
> Here is the PDF version of our responses: https://www.dropbox.com/s/0r2hztg7af87934/Response_1_ICLR_2019.pdf?dl=0  (anonymous link)
>
> The authors appreciate the reviewer’s the time and efforts for reviewing this paper, and would like to respond to the questions in the following paragraphs.
>
> Q1: The paper proposes an exploration strategy for deep reinforcement learning agent in continuous action spaces.
>
> Response: We would like to sincerely apologize for the misunderstanding caused.  In fact, our adversarial exploration strategy is a self-supervised data-collection strategy developed for training an inverse dynamics model, which is formally described in Sections 2.2 and 3.  Different from exploration strategies for deep reinforcement learning (DRL) (which aim at learning policies for maximizing the expected returns in RL tasks), our work focuses on discovering a policy for collecting a useful training dataset for an inverse dynamics model.
>
> Q2: Seems unstable and not clear how it would scale in a large state space where most states are going to be very difficult to learn about in the beginning like a humanoid body.
>
> Response: We would like to address the reviewer’s concerns in the following two paragraphs.
>
> First, with regard to the “instability” issue, we are not quite sure which aspect the reviewer refers to, and would appreciate it if the reviewer could kindly share some more information with us.  We assume that the reviewer could be referring to either the variance of the training losses, or the variance in the learning curves of Fig. 3.  For the former case, a stabilization technique is presented in Section 3.3, and its effectiveness is analyzed and validated in Section 4.5.  For the latter case, the variance in the learning curves is mainly caused by the complexity of the high-dimensional observation spaces.  As contemporary DRL techniques also suffer from the same problems when training with raw images (i.e., high-dimensional observation spaces) [1], the high variance in the learning curves can similarly be alleviated by enhancing the model architecture or the training algorithm.  Please note that this issue also occurs in the learning curves of the other baseline methods.  Discussion of model architectures and specific training techniques for high-dimensional observation spaces, however, is beyond the scope of this paper.
>
> Second, we agree with the reviewer that learning a policy in an environment with a large state space has been a challenging research topic.  However, in the past few years, a number of DRL methods have been proposed and achieved remarkable successes in such environments, including humanoid robotic control [2].  The successes of these methods indicate that even in an environment with a large state space, it is still possible to discover an effective policy that maximizes the expected return.  In the proposed adversarial exploration strategy, we train a DRL agent to maximize the expected losses of the inverse dynamics model (Eq. (6)).  As DRL methods have been shown effective in exploiting arbitrary reward functions in large state spaces in the literature, we consider that our method can be extended to environments with large state spaces, and exploit the losses of the inverse dynamics model to collect training samples accordingly.

---

> > ### Author Response · Authors · 2018-11-16
> > **Response to reviewer 1 (Part 2/2)**
> >
> > Q3: Only accounts for the immediately controllable aspects of the environment which doesn't seem to be the hard part. Understanding the rest of the environment and its relationship to the controllable part of the state seems beyond the scope of this model.
> >
> > Response: The authors would like to thank the reviewer for raising this question, and would love to address the reviewer’s concern by clarifying it in two aspects.  First, training a satisfactory inverse dynamics model for the controllable part of the environment is essentially not straightforward.  As the comparisons presented in Section 4 reveal, there exist large variations in the success rates of different self-supervised data-collection strategies.  For example, in FetchPickAndPlace of Fig. 3, there exists a significant performance gap between our method and the “Random” baseline, indicating that different exploration strategies do lead to different learning curves.  A simple exploration strategy (e.g.,  Random) might not be as effective as the other strategies in some tasks (e.g., FetchPush, FetchPickAndPlace, and HandReach).  Second, a key feature of our adversarial exploration strategy is that it utilizes a DRL agent to keep exploring the environment and collecting difficult training samples for the inverse dynamics model.  Therefore, our method allows the inverse dynamics model to gradually expand the scope of the controllable environment.
> >
> > Q4: Nonetheless I can imagine it helping with initial random motions.
> >
> > Response: We are afraid that there seems to be some misunderstanding, and are not quite sure if we understand the reviewer’s meaning of “helping with initial random motions” correctly.  We were just wondering if the reviewer could kindly clarify this question for us?
> >
> > Q5: From Eq. (6) the bonus seems to be unbounded and Eq. (7) doesn't seem to fix that. Is that not an issue in general ? Any intuition about that ?
> >
> > Response: Although most of the DRL works suggest that the rewards should be re-scaled or clipped within a range (e.g., from -1 to 1), the unbounded rewards do not introduce any issue during the training process of our experiments.  The empirical rationale is that the rewards received by the DRL agent are regulated by Eq. (7) to be around δ, as described in Section 4.5 and depicted in Fig. 4.  Without the stabilization technique, however, the learning curves of the inverse dynamics model degrade drastically (as illustrated in Fig. 5), even if the reward clipping technique is applied.
> >
> > [1] M. Fortunato et al., "Noisy networks for exploration," in Proc. Int. Conf. Learning Representations (ICLR), Apr.-May, 2018.
> >
> > [2] Ł. Kidziński et al., "Learning to run challenge solutions: Adapting reinforcement learning methods for neuromusculoskeletal environments," arXiv:1804.00361, Apr. 2018.

---

### Official Review · AnonReviewer2 · 2018-11-03
**good training exploration, somewhat limited scope of experimental conditions**

**Rating:** 7
**Confidence:** 3

**Review:**

This paper presents a system for self-supervised imitation learning using a RL agent that is rewarded for finding actions that the system does not yet predict well given the current state.  More precisely, an imitation learner I is trained to predict an action A given a desired observation state transition xt->xt+1; the training samples for I are generated using a RL policy that yields an action A to train given xt (a physics engine evaluates xt+1 from xt and A).  The RL policy is rewarded using the loss incurred by I's prediction of A, so that moderately high loss values produce highest reward.  In this way, the RL agent learns to produce effective training samples that are not too easy or hard for the learner.  The method is evaluated on five block manipulation tasks, comparing to training samples generated by other recent self-supervised methods, as well as those found using a pretrained expert model for each task.

Overall, this method exploration seems quite effective on the tasks evaluated.  I'd be curious to know more about the limits and failures of the method, e.g. in other types of environments.

Additional questions:

- p.2 mentions that the environments "are intentionally selected by us for evaluating the performance of inverse dynamics model, as each of them allows only a very limited set of chained actions".  What sort of environments would be less well fit?  Are there any failure cases of this method where other baselines perform better?

- sec 4.3 notes that the self-supervised methods are pre-trained using 30k random samples before switching to the exploration policy, but in Fig 2, the success rates do not coincide between the systems and the random baseline, at either samples=0 or samples=30k --- should they?  if not, what differences caused this?

- figs. 4, 5 and 6 all relate to the stabilizer value delta, and I have a couple questions here:  (i) for what delta does performance start to degrade?  At delta=inf, I think it should be the same as no stabilizer, while at delta=0 is the exact opposite reward (i.e. negative loss, easy samples).  (ii) delta=3 is evaluated, and performance looks decent for this in fig 6 --- but fig 4 shows that the peak PDF of "no stabilizer" is around 3 as well, yet "no stabilizer" performs poorly in Fig 5.  Why is this, if it tends to produce actions with loss around 3 in both cases?

---

> ### Author Response · Authors · 2018-11-16
> **Response to reviewer 2 (Part 1/2)**
>
> Here is the PDF version of our responses: https://www.dropbox.com/s/707hka5ba9abg54/Response_2_ICLR_2019.pdf?dl=0 (anonymous link)
>
> The authors appreciate the reviewer’s the time and efforts for reviewing this paper, and would like to respond to the questions in the following paragraphs.
>
> Q1: Overall, this method exploration seems quite effective on the tasks evaluated.  I'd be curious to know more about the limits and failures of the method, e.g., in other types of environments.
>
> Response: We would like to bring to the reviewer’s kind attention that although our method outperforms  all the baseline methods in most of the tasks, all the methods (including ours) are unable to surpass the “Demo” baseline in the HandReach task.  The observation implies that with regard to inverse dynamics model training, the contemporary self-supervised data-collection strategies (including ours and the baseline methods) are not as effective as training directly with expert demonstrations for this task.  We consider that this task is a typical failure case for our method (as well as the other baseline methods).   The underlying rationale is presumably due to the difficulty for exploration in the high-dimensional action space of HandReach, as this is the major difference between it and the other tasks presented in our paper.  We have included more analyses on the limitations and failure cases of our method in Section 4.3 of the revised version.
>
> Q2: p.2 mentions that the environments "are intentionally selected by us for evaluating the performance of inverse dynamics model, as each of them allows only a very limited set of chained actions".  What sort of environments would be less well fit?  Are there any failure cases of this method where other baselines perform better?
>
> Response: We would like to thank the reviewer for raising this question.  In fact, environments that allow various valid actions for a given transition (x_t to x_{t+1}) would be less well fit for our method.  As we train the inverse dynamic model by minimizing mean-square error between the predicted action a and the ground truth action â (Eq. (5)), multiple ground truth actions for the same transition would lead to high variance in the derived gradients.  This is referred to as the “multimodality problem” and has been discussed in [1].  As the main focus of this paper is to investigate the effectiveness of the proposed adversarial exploration strategy for self-supervised imitation learning, we do not incorporate these environments and the multimodality problem in our scope of discussion to avoid confusion and potential distraction of the main subject.
>
> Q3: Sec 4.3 notes that the self-supervised methods are pre-trained using 30k random samples before switching to the exploration policy, but in Fig 2, the success rates do not coincide between the systems and the random baseline, at either samples=0 or samples=30k --- should they?  if not, what differences caused this?
>
> Response: We would like to sincerely apologize for the misunderstanding caused.  In our experiments, pre-training with random samples is only applied to the HandReach task due to its high complexity in exploration.  This is also the primary subject that we intend to discuss in Section 4.3.  As a result, the success rates of all the methods are the same in the HandReach task for the first 30K samples, as you have noticed.  For the other tasks, we do not pre-train the models with random samples.  Therefore, their success rates before 30K do not coincide with that of the “Random” baseline.

---

> > ### Author Response · Authors · 2018-11-16
> > **Response to reviewer 2 (Part 2/2)**
> >
> > Q4: Figs. 4, 5, and 6 all relate to the stabilizer value delta, and I have a couple questions here:  (i) for what delta does performance start to degrade?  At delta=inf, I think it should be the same as no stabilizer, while at delta=0 is the exact opposite reward (i.e. negative loss, easy samples).  (ii) delta=3 is evaluated, and performance looks decent for this in fig 6 --- but fig 4 shows that the peak PDF of "no stabilizer" is around 3 as well, yet "no stabilizer" performs poorly in Fig 5.  Why is this, if it tends to produce actions with loss around 3 in both cases?
> >
> > Response: (i) Many thanks for raising this interesting question.  We have conducted additional experiments to investigate this issue, and have analyzed the results in the updated version of our manuscript.  In Fig. 6, we compare the learning curves of the imitator for different values of δ.  For instance, Ours(0.1) corresponds to δ = 0.1.  It is observed that for most of the tasks, the success rates drop when δ is set to an overly high or low value (e.g., 100.0 or 0.0), suggesting that a moderate value of δ is necessary for the stabilization technique.  The value of δ can be adjusted dynamically by the adaptive scaling technique presented in [2], which is left as our future direction.
> >
> > (ii) We would like to clarify this issue as follows.  Although the peaks of “Ours(w/o stab)” in Fig. 4 are around 3, it does not suggest that their final success rates are comparable to those of “Ours(3.0)” (with our stabilization technique) in Fig. 6.  Please note that Fig. 4 only plots the first 2K training batches of the inverse dynamics model in the entire training process.  After 2K, the peaks of “Ours(3.0)” and “Ours(w stab)” still stay close to their $\delta$ values for the rest of the training processes, while that of “Ours(w/o stab)” gradually grows to around 1K, which is prohibitively higher than reasonable values.  Such a high training loss could cause the gradient exploding problem, which typically leads to a severe performance drop [3].  This also explains why “Ours (w/o stab)” performs poorly in Fig. 5.  As a result, “Ours(3.0)” is superior to “Ours (w/o stab)” due to its relatively stabler gradients.
> >
> > Please also note that plotting the training losses of only the first 2K training batches in Fig. 4 is intended for enhancing the visualization and readability of the results.  As the training losses of  ”Ours(w/o stab)” span from low values to extraordinarily high values, it is not feasible to be compare the PDF of ”Ours(w/o stab)” directly with that of “Ours(w stab)”.  Therefore, we selected a range of data (i.e., the first 2K batches) in which the training losses of “Ours(w/o stab)” are still under 10.  We have incorporated additional figures in our supplementary material for illustrating the above observations.
> >
> > [1] P. Deepak et al., "Zero-shot visual imitation," in Proc. Int. Conf. Learning Representations (ICLR), Apr.-May 2018.
> >
> > [2] M. Plappert et al., "Parameter space noise for exploration." in Proc. Int. Conf. Learning Representations (ICLR), Apr.-May, 2018.
> >
> > [3] R. Pascanu, T. Mikolov, and Y. Bengio, "On the difficulty of training recurrent neural networks," in Proc. Int. Conf. Machine Learning (ICML), pp.1310-1318, Jun. 2013.

---

### Meta-Review · Area_Chair1 · 2018-12-16

**Confidence:** 4
**Recommendation:** Reject

**Metareview:**

This paper proposes a method for incentivizing exploration in self-supervised learning using an inverse model, and then uses the learned inverse model for imitating an expert demonstration. The approach of incentivizing the agent to visit transitions where a learned model performs poorly. This relates to prior work (e.g. [1]), but using an inverse model instead of a forward model. The results are promising on challenging problem domains, and the method is simple. The authors have addressed several of the reviewer concerns throughout the discussion period.
However, three primary concerns remain:
(A) First and foremost: There has been confusion about the problem setting and the comparisons. I think these confusions have stemmed from the writing in the paper not being sufficiently clear. First, it should be made clear in the plots that the "Demos" comparison is akin to an oracle. Second, the difference between self-supervised imitation learning (IL) and traditional IL needs to be spelled out more clearly in the paper. Given that self-supervised imitation learning is not a previously established term, the problem statement needs to be clearly and formally described (and without relying heavily on prior papers). Further, the term self-supervised imitation learning does not seem to be an appropriate term, since imitation learning from an expert is, by definition, not self-supervised, as it involves supervisory information from an expert. Changing this term and clearly defining the problem would likely lead to less confusion about the method and the relevant comparisons.
(B) The "Demos" comparison is meant as an upper bound on the performance of this particular approach. However, it is also important to understand what the upper bound is on these problems in general, irrespective of whether or not an inverse model is used. Training a policy with behavior cloning on demonstrations with many (s,a) pairs would be able to better provide such a comparison.
(C) Inverse models inherently model the part of the environment that is directly controllable (e.g. the robot arm), and often do not effectively model other aspects of the environment that are only indirectly controllable (e.g. the objects). If the method overcomes this issue, then that should be discussed in the paper. Otherwise, the limitation should be outlined and discussed in more detail, including text that outlines which forms of problems and environments this approach is expected to be able to handle, and which of those it cannot handle.

Generally, this paper is quite borderline, as indicated by the reviewer's scores. After going through the reviews and parts of the paper in detail, I am inclined to recommend reject as I think the above concerns do not outweigh the pros.

One more minor comment is that the paper should consider mentioning the related work by Torabi et al. [2], which considers a similar approach in a slightly different problem setting.

[1] Stadie et al. https://arxiv.org/abs/1507.00814
[2] Torabi et al. IJCAI '18 (https://arxiv.org/abs/1805.01954)